# Optimization of Heat-Resistance Technology for a Duck Hepatitis Lyophilized Live Vaccine

**DOI:** 10.3390/vaccines10020269

**Published:** 2022-02-10

**Authors:** Yanhong Zhao, Bihua Deng, Xiaoqing Pan, Jinqiu Zhang, Xiaoxin Zuo, Junning Wang, Fang Lv, Yu Lu, Jibo Hou

**Affiliations:** 1Institute of Veterinary Immunology & Engineering, National Research Center of Engineering and Technology for Veterinary Biologicals, Jiangsu Academy of Agricultural Sciences, Nanjing 210014, China; Zhaoyh20090713@163.com (Y.Z.); dengbihua2000@163.com (B.D.); jqzh@126.com (J.Z.); zxx0086@126.com (X.Z.); wjn_2016@126.com (J.W.); houjiboccvv@163.com (J.H.); 2Livestock Research Institute, Jiangsu Academy of Agricultural Sciences, Nanjing 210014, China; pxq1611@jaas.ac.cn; 3Jiangsu Co-Innovation Center for the Prevention and Control of Important Animal Infectious Disease and Zoonose, Yangzhou University, Yangzhou 225009, China; 4Jiangsu Key Laboratory for Food Quality and Safety-State Key Laboratory Cultivation Base, Ministry of Science and Technology, Nanjing 210014, China; 5School of Pharmacy, Jiangsu University, Zhenjiang 212013, China

**Keywords:** DHV, optimization of the freeze-drying process, stabilizers, thermostability

## Abstract

In this study, to improve the quality of a live attenuated vaccine for duck viral hepatitis (DHV), the lyophilization of a heat-resistant duck hepatitis virus vaccine was optimized. The optimized heat protectors were made of 10% sucrose, 1.2% pullulan, 0.5% PVP, and 1% arginine, etc., with a titer freeze-drying loss of ≤0.50 Lg. The vaccine product’s valence measurements demonstrated the following: the vaccine could be stored at 2–8 °C for 18 months with a virus titer loss ≤0.91 Lg; at 37 °C for 10 days with a virus valence loss ≤0.89 Lg; and at 45 °C for 3 days with a virus titer loss ≤0.90 Lg. Regarding safety, no deaths occurred in two-day-old ducklings immunized with a 10 times dose vaccine; their energy, diet, and weight gain were all normal, demonstrating that the DHV heat-resistant vaccines were safe for ducklings and did not cause any immune side effects. Duck viral hepatitis freeze-dried vaccine began to produce antibodies at 7 d after immunization, reached above 5.0 on 14 d, and reached above 7.0 on 21 d, showing a continuous upward trend. This indicates that duck viral hepatitis vaccine has a good immunogen level. The optimization of the freeze-drying process saves costs and also improves the quality of the freeze-drying products, which provides important theoretical and technical support for the further study of vaccine products.

## 1. Introduction

Duck viral hepatitis (DHV) is an acute, highly contagious, and deadly infectious disease caused by the duck hepatitis virus (DHV), which mostly occurs in ducklings around three weeks of age. The clinical symptoms of diseased ducklings are opisthotonos and liver lesions. DHVs can be classified into duck hepatitis A viruses (DHAVs), of the Picornaviridae family and Avihepatovirus genus, and duck astroviruses (DAst Vs), of the Astroviridae family and Avastrovirus genus. DHAVs are divided into three serotypes: type 1, type 2, and type 3 [1,2]. DHAVs account for more than 80% of the mortalities in ducklings less than 21 days old that are raised on duck farms, and this virus is prevalent worldwide [3]. To date, through phylogenetic analyses and neutralization tests, DHAVs have been classified into three serotypes: DHAV-1, DHAV-2, and DHAV-3. DHAV infection was first described in the USA, in 1945, and the virus was first isolated in 1950 [4,5]. DHAV-1 is the classical and most widespread serotype [6]. DHAV-2 is only endemic in Taiwan [7]. DHAV-3 accounts for an increasing proportion of DHAV cases in South Korea [8], China, and Vietnam [9]. The use of DHAV live attenuated vaccines around the world has provided reliable protection for vaccinated ducklings [10,11,12,13,14]. Therefore, it is extremely important to develop high-quality vaccine products.

Generally, domestic veterinary live vaccines have been mostly prepared by the traditional freeze-drying process, mainly using domestic sucrose, gelatin, skim milk, and other simple groups, with the freeze-dried vaccine usually stored at −15 °C; however, their stability can still be affected. For approximately 30–40% of protein formulations, including three of the ten best-selling drug products worldwide in 2016, at least one formulation is freeze-dried for simulation and reconstituted before clinical use (Remicade, Enbrel, and Herceptin). A low processing temperature does not induce the severe degradation of proteins that occurs during drying by other methods [15]. Foreign oligosaccharides, polyols, protein, antioxidants, buffers, fillers, and other composites that have prescription compatibility, in freeze-dried vaccines prepared at 2–8 °C, can be stored for 2 years [16]. In fact, although the application of freeze-drying technology has a long history, in actual freeze-dried formulations, most of the freeze protectants still use gelatin; gelatin is a macromolecular protein, originating from animal tissues, and is also the main source of bacterial endotoxin. In the case of 21 soldiers with allergic reactions after rabies vaccination, the German doctor Mylene Niclou found that gelatin was the main cause of severe allergy after rabies vaccine injection [17]. There is extensive literature that supports the idea that the gelatin composition is the main cause of allergic reactions after vaccinations [18,19]. Newly developed protectants use non-animal-derived excipients rather than gelatin components. Chun [20,21] aimed to replace gelatin as a stabilizer for a varicella virus liquid vaccine using a heat-resistant protective agent and freeze-drying as a complementary process. At this time, the freeze-drying process is also particularly important. In the prefreezing method, unreasonable prefreezing and drying temperatures can easily lead to complications from freeze-drying, such as collapse, shrinkage, and melting, and even significantly reduce the cold titer of the dry vaccine.

Pharmaceutical preparations have long been based on the thermal parameters of drugs to determine the prefreeze and drying temperatures, and then further scientific optimization of the freeze-drying procedures. The thermolabile nature of commercially available vaccines necessitates their storage, transportation, and dissemination under refrigerated conditions. The maintenance of a continuous cold chain at every step increases the final costs of vaccines. Any breach in the cold chain, even for a short duration, requires the vaccines to be discarded. Therefore, there is a pressing need for the development of thermostable vaccines [22]. Therefore, the aim of this study was to design a heat-resistant protective agent for sucrose, pullulan, PVP, and arginine, according to the eutectic point of the formulation, using scientifically determined temperatures in freeze-drying procedures to accelerate an anti-aging test with 37 and 45 °C residual water for screening indicators. We obtained a good heat-resistant composition and freeze-drying process, with a view to using suitable temperatures for animal vaccines for the development of new formulations and to provide technical support.

## 2. Materials and Methods

### 2.1. Materials

The duck viral hepatitis type 1 DHV-JS strain was isolated in our laboratory (microbial preservation number CGMCC no. 8159) [23]. In the experiment, the duck embryos were used at 11–12 days of gestation, and ducklings were used at 1–7 days of age. All the ducklings were from healthy Cherry Valley ducks without a history of duck hepatitis and with negative valences for anti-DHV antibodies.

### 2.2. Instruments

The instruments used included a 0.15 M^2^ freeze dryer (Advantage Plus, VirTis, ID, 815 Route 208 Gardiner, New York, NY, USA), a DSC thermal analyzer (PerkinElmer, Waltham, MA, USA), a Scaning Electron Microscope(SEM) (FEI, Hillsboro, OR, USA), and 37 °C cell incubator (Thermo, 81 Wyman Street, Waltham, MA, USA).

### 2.3. Vaccine Lyophilizatio

The DHV liquid was prepared as described in the literature [23]. For the protection of the DHV formulation during lyophilization, eight formulations of protective agents were prepared using different proportions of sucrose, Pullulan, PVP, and arginine (Table 1). Filtration sterilization was used for the heat-labile components, while high-temperature, high-pressure sterilization was used for the heat-resistant components [24]. The eight formulations of the agents were mixed with the liquid virus from the duck viral hepatitis at a ratio of 1:1 (*v/v*), and then, the mixtures were aliquoted into 7 mL vials (2 mL in each vial). After partial sealing using a butyl stopper, the vials were placed in the cabinet in the freeze dryer and freeze-dried. Subsequently, preliminary screening of the heat-resistant protective agents was conducted based on comparisons of indicators such as the frozen form, color, texture, moisture content, vacuum degree, and virus loss before and after freeze-drying. A rapid freeze-drying process was used (Figure 1).

### 2.4. Accelerated Thermal Stability Test

The freeze-dried products were stored at 37 °C for 10 days, and then, the shapes of the products were observed. The ELD_50_ residual moisture and vacuum degree of the viral titers were measured for the vaccines as previously described. A difference in the decrease in viral titers equal to or less than 1.0 Lg was set as the standard for the screening of the formulations [25].

### 2.5. Measurement of Thermal Parameters for the Heat-Resistant Formulation

The final vaccine freeze-drying process, described in Section 2.3, was screened using the accelerated thermal stability test, described in Section 2.4. The formulation with the best heat resistance was chosen, and then, the eutectic point and the collapse temperature for the formulation were measured. The eutectic point was measured using the electrical resistance, and the collapse temperature was measured using a low-temperature freeze-drying microscope. The conditions for the measurement were as follows: temperature, −40 °C; vacuum, 20.0 Pa; ramp row, 3; rate, 3 °C/min; and limit, −40 °C. The test was conducted by the Toffion Science and Technology Group Company Limited (NO. 1509. Duhui Road. Minhang District, Shanghai, China.

### 2.6. Vaccine Heat-Stability Test

The screened protective agent A7 was mixed well with duck viral hepatitis, and then, the mixture was repeatedly freeze-dried into three batches, according to the optimized procedure. The physical properties, vacuum degree, and residual water after freeze-drying were measured for the three batches of vaccines, consistent with the Veterinary Pharmacopoeia of People’s Republic of China [26]. The products were stored at 2–8, 37, and 45 °C, and the shapes of the products were then observed [27]. The viral titers, ELD_50_ residual moisture, and vacuum degree were measured for the vaccines as previously described. A difference in the decrease in viral titers equal to or less than 1.0 Lg was set as the standard for the screening of the formulations. At the same time, forms of 37 and 45 °C protection on the market were assessed to compare the thermal resistance.

### 2.7. Volumetric Karl Fischer (Moisture)

The moisture measurement of the lyophilized cake was performed with a DL31 Volumetric Karl Fischer Titrator (Mettler Toledo, Columbus, OH, USA) on a Mitsubishi, Model RV-2AJ-511, TIX Robotic Titration System (AB Controls, Inc., Irvine, CA, USA). The samples were crushed in a glove box with a relative humidity <3% and loaded into tared tubes. A sample size of approximately 100 mg was used. The tubes were automatically loaded and titrated with Hydranal. Standards were run at the start of analysis, after every 4 samples, and at the end of analysis. The titrator was washed with methanol between samples [28].

### 2.8. Scanning Electron Microscopy

Imaging was performed with a Quanta 3D FEG scanning electron microscope (SEM) (FEI, Hillsboro, OR, USA). A core plug of the lyophilized cake in the vertical axis was taken using a 3 mm-diameter borer. The sample plug was laid on its side (to expose the entire depth of the cake) and affixed on aluminum stubs with carbon adhesive tape. The SEM images were collected at 5 kV [29].

### 2.9. Safety Analysis of the Heat-Resistant DHVs

The safety analysis of heat-resistant DHVs was conducted as previously described in Section 2.1 [23]. Two-day-old ducklings were selected. The ducklings in the experimental group were immunized by heat-resistant duck viral hepatitis vaccines through two approaches (nasal drops/eye dropping and intramuscular injection) for a 10 times dose. There were 20 ducklings in each group, and immunization was not performed for the control group. After inoculation, these ducklings were quarantined and observed for 14 days. The clinical conditions of the ducklings, including energy, food and water intake, growth, and weight gain, were observed after vaccine inoculation.

### 2.10. Immunogenicity Test of DHV Freeze-Dried Vaccines

The immunogenicity test was divided into 3 groups, with 7-day-old ducklings in each group that were subcutaneously injected into the neck, single immunization, and the blood collected on 7 d, 14 d, and 21 d after immunization, and the serum was separated. After mixing by volume, the duck embryo neutralization test was used to determine the level of duck viral hepatitis antibody in the duck blood. The specific test groups are shown in Table 2.

## 3. Results

### 3.1. Screening of DHV Heat-Resistant Protective Agents

The virus titer losses were all less than 1.0 Lg after the storage of the freeze-dried vaccines prepared using formulations A1–A8 at 37 °C for 10 days. The formulation with the most loss was A7, where the lyophilized loss was 0.403 Lg; the heat loss of each stabilizer formulation was 0.313 Lg (Figure 2).

### 3.2. Vaccine Heat-Stability Test

The three batches of vaccines that had been repeatedly freeze-dried (see Section 2.6) were stored at 2–8 °C for up to 18 months, at 37 °C for 14 d, and at 45 °C for 3 days; a 1.0 Lg loss of virus titers was considered the endpoint of vaccine storage (Figure 3a–c). The results show that the loss of viral load for the three batches of live DHV A7 was 0.53–0.75 Lg after storing at 2–8 °C for 18 months, showing effective protection against a loss of duck viral hepatitis virus. At 37 and 45 °C, the titer of the virus valence was better than that of commercial seedlings.

### 3.3. Measurement of Thermal Parameters for Heat-Resistant Formulations

Formulation A7, which showed the best heat resistance, was screened as described in Section 2.1, and the thermal parameters were measured (Table 3).

### 3.4. Relevant Physical and Chemical Test Results for Products Prepared with the Heat-Resistant Freeze-Dried Protective Agent

The physical properties, vacuum degree, and residual (vacuum detection adopts a high-frequency spark vacuum measurement) water after freeze-drying (volumetric Karl Fischer, Mettler Toledo, Zurich, Switzerland) were measured for the three batches of vaccines (Table 4), consistent with the Veterinary Pharmacopoeia of People’s Republic of China.

### 3.5. Scanning Electron Microscopy

The SEM detection of the DHV commodity seedlings and the heat-resistant freeze-dried vaccine showed that the group of samples mainly had a porous honeycomb structure. In the b group, the samples’ porous honeycomb structure was more uniform and regular. Therefore, the sublimation rate of the bound water was faster. In clinical redissolution, dilutions can quickly penetrate into small pores, enabling it to quickly dissolve and restore activity (Figure 4).

### 3.6. Safety of Live DHVs Vaccine

Animal safety experiments were performed using the freeze-dried vaccine with formulation A7. There were three groups in this experiment. After immunization, quarantine observations were performed for 14 days [24], and the following clinical conditions of the vaccinated ducklings were observed: energy, food intake, water intake, growth, and weight gain. As shown in Table 4, no immunized ducklings died after the injection of a large dose using two immunization approaches (10 times the recommended immunization dose). The comparisons between the control group and immunization group showed that there were no significant differences in weight gain in the ducks (*p* > 0.05) (Table 5).

### 3.7. Immunogenicity Test of DHV Freeze-Dried Vaccines

The antibody levels 7 d, 14 d, and 21 d after immunization with duck viral hepatitis freeze-dried vaccine are shown in the Figure 5. Antibodies began to be produced 7 d after immunization and reached above 5.0 on 14 d and above 7.0 on 21 d, showing a continuous increase. The results from the figure show the neutralizing antibodies of the freeze-dried vaccine and the commercial vaccine produce effective antibodies to ducks, and the antibodies of the freeze-dried vaccine are higher than those of the commercial vaccine, indicating that the duck viral hepatitis vaccine we have studied has a good immunogen level.

## 4. Discussion

Freeze drying is widely used in pharmaceutical industry to stabilize labile drugs by solvent sublimation at low pressure; it encompasses three different stages: freezing, primary drying, and secondary drying [30]. Various methods of prefreezing and different sizes of ice crystal particles can cause varying levels of mechanical damage to microorganisms. Quick freezing can cause the formation of fine ice-crystal particles, whereas slow freezing can lead to the formation of large ice crystals, and annealing can cause the secondary crystallization of the products, leading to an increase in the glass transition temperature of the freeze-dried products [31,32]. Therefore, in this study, we designed a freeze-drying formulation by measuring the thermal parameters of several formulations. The chemical stability of drug and food components during freeze-drying and storage is often influenced by factors such as the chemical composition, freeze-drying rate, freezing, preservation of ambient temperature and humidity, and glass transition temperature (Tg) [33,34]. When a vaccine has a Tg temperature that is about 40 °C above the target preservation temperature, once the ambient temperature of the vaccine product is higher than its own Tg, the vaccine product will exhibit a series of trait changes such as collapse, hardening, and discoloration. Due to the damage of the original loose structure, the rehydration of the product is seriously slowed; therefore, the selection of a protective agent that can improve the product’s Tg is conducive to the stability of the product in the production and preservation processes.

In the development of live attenuated vaccines, some chemical and physical reactions can easily destroy the structure of the virus surface proteins and internal nucleic acids [35]. Before vacuum drying, freezing damage is an important reason for the decline in food and drug quality. In addition, during the storage process for freeze-drying materials, protein condensation, oxidation reactions, and hydrolysis can appear in the materials, and these phenomena lead to the degeneration of the active components during freeze-drying and storage [36]. The addition of protective agents can improve the stability of the vaccine and extend its shelf life.

Lyophilized protectants can protect the activity of biological products and also act as excipients. Protective agents inhibit protein aggregation, reduce surface adsorption, and play a role in skeleton support. Common excipients include polyvinyl pyrrolidone (PVP), polyethylene glycol, albumin, gelatin, defatted milk, and other polymers. The effects of these excipients on protein stability in solution are mainly due to their interactions with the protein and, most importantly, with water. In a dry state, the excipients provide a stable environment for proteins through their interactions with water [37]. However, vaccines produced with gelatin-containing protective agents are at risk of causing human allergic reactions [38]. Some gelatin-free protective agents are used in live attenuated human vaccines for encephalitis and measles, mumps, and rubella combined [39]. It is necessary to accelerate the development of gelatin-free vaccines in animals, and this study of gelatin-free thermal protectors has yielded good results.

In the present study, a protective formulation for ensuring heat stability consisting of 10% sucrose,1.2% pullulan, 0.5% PVP, and 1% arginine showed good protective effects and safety in ducks. Sucrose prevents the secondary structure of the protein becoming altered during freeze-drying, preventing protein denaturation during freeze-drying, and allowing the vaccine to retain its structural and functional integrity despite the water loss from freezing and drying [40]. Pullulan, an exocellular homopolysaccharide produced by Aureobasidium pullulans, is a linear mixed linkage α-D-glucan consisting mainly of maltotriose repeating units interconnected by α-(1→6) linkages. The regular alternation of α-(1→4) and α-(1→6) glycosidic bonds in pullulan results in its structural flexibility and enhanced solubility. Pullulan interacts with starch or protein and plays an important role in the network of them [41]. PVP is a great stabilizer, preventing the aggregation of NPs via the repulsive forces that arise from its hydrophobic carbon. chains that extend into solvents and interact with each other (steric hindrance effect) [42]. Arginine became the preferred choice for the amino acid composition in the protectant formulation due to its unique chemical structure, resulting in an increase in the solubility of the protein long before freezing, showing protective effects, and reducing aggregation in solution [43]. Adding a certain amount of arginine as a lyophilized protective agent for pig-derived fibrinogen improved the quality of all aspects of the products. This may be because, in the subsequent dehydration process, when the amino acid can form hydrogen bonds and special sites on the protein surface to provide thermodynamic stability for protein molecules, one arginine group has the potential to form seven side-chain hydrogen bonds, while histidine only has the potential to form three side-chain hydrogen bonds. In addition, arginine does not exhibit water resistance, and its eutectic point and glass transition temperature are high relative to that of histidine; therefore, the effects of arginine in lyophilization are strong.

The stability of the protein is easily affected by the acidity and alkalinity of its environment, its stability in solution, and its liability to conformational changes due to various stresses encountered during purification processing and storage [44]. The freeze-drying stability is measured by the change in virus titer before and after freeze-drying, and the heat resistance is the attenuation of the vaccine sample content by storage at 37 °C; these are used to judge the stability of biological products. The traditional viral titer determination method, ELD_50_, takes 1 week, and the method itself can also have errors, which can easily cause misleading results. Finding new, fast, and accurate assays is a challenge in vaccine lyophilization, such as differential scanning calorimetry, near-infrared spectrometry, and scanning electron mirrors, which play an important role in the early stage for quickly judging the glass transition temperature of a protective agent to help judge its protective effect. In this study, we combined the traditional ELD_50_ detection with the Tg, SEM, and the 45 °C detection index, to accelerate the screening of the preliminary formulations and the detection of the results, therefore realizing a quick and accurate detection method for vaccine product detection.

The results show that the freeze-dried products could be stored at 2–8 °C for 18 months, with virus loss as small as 0.91 Lg, demonstrating good reproducibility. Accelerated age-resistance tests for the freeze-dried products after storage at 37 °C for 14 days indicated that the virus loss was only 0.89 Lg; for storage at 45 °C for 3 days, the virus loss was only 0.90 Lg. Two-day-old ducklings were inoculated with 10 doses of the vaccine through nasal drops/eye drops or intramuscular injection for 14 days, and all the ducklings remained healthy and produced effective antibody protection.

Thus, our findings provide insights into optimal heat-resistant protective agents for use in the preparation of DHVs and establish a scientifically rational freeze-drying technique. Moreover, we optimized the prefreezing method and provide important data to support the choice of freeze-drying techniques for heat-resistant veterinary vaccines. The lyophilized DHV vaccine may facilitate easier application for duckling farmers, since the vaccine can be conveniently stored and transported, unlike previous vaccines that required live frozen storage.

## 5. Conclusions

In this study, we studied the functioning of a frozen duck viral hepatitis vaccine used with no gelatin thermal protective agent. Combined with the 45 °C test indicators, Tg, SEM, and screening sped up the preliminary formulation, thus being an accurate and rapid detection method; this was also due to the best product freeze-drying technology, as taken from the correlation test. Lyophilized heat-resisting protective agent, 10% sucrose, 1.2% pullulan, 0.5% PVP, and 1% arginine, through the rapid freeze-drying process, can make the vaccine products be stored under 2–8 °C for 18 months, at 37 °C protection for 14 days and 45 °C protection for three days. The heat-resistance formula of gelatin effectively solved the problem of animal vaccine products caused by adding exogenous gelatin allergic disease.

This study in order to speed up the development of duck virus hepatitis vaccine provides an important theoretical support. Above all, heat protectant A7 and freeze-drying technology can provide effective protection for DHV-3. Not only does gelatin-free vaccine research provides theoretical support, it would also lead to reducing the production costs. Although we have conducted laboratory research, mass production still needs further research.

## Figures and Tables

**Figure 1 vaccines-10-00269-f001:**
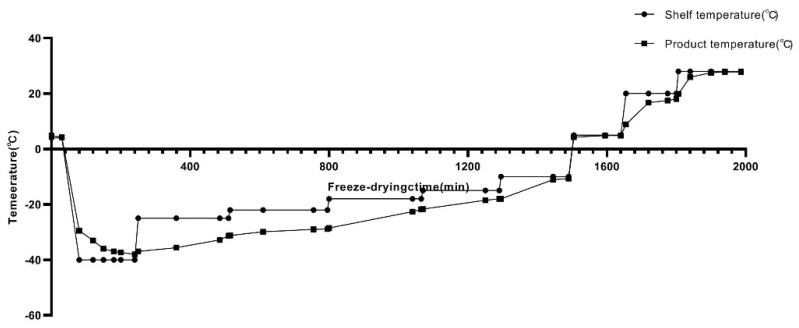
The rapid freeze-drying process. Each formulation was mixed 1:1 with a DHV antigen, and then dispended into 7 mL sterile glass vials. The lyophilized procedures were set as below. Samples were frozen to −40 °C at a rate of 1.13 °C/min. Primary drying was performed at −25 °C. Secondary drying was performed at 28 °C. Vials were rubber-stoppered under vacuum and tightly sealed with an aluminum cap under normal air pressure.

**Figure 2 vaccines-10-00269-f002:**
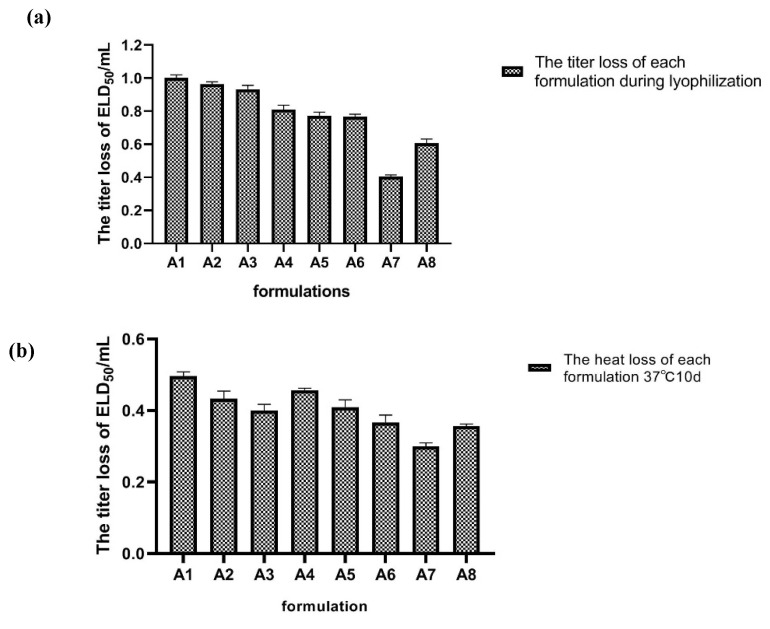
Screening results for the stabilizer formulations. (**a**) The titer loss of each stabilizer formulation during lyophilization. The lyophilization loss is the difference between the vaccine titers before and after the lyophilization of the formulations. (**b**) The heat loss of each stabilizer formulation. The heat loss was the titer reductions for the lyophilized stabilizer formulations before and after incubating at 37 °C for 10 days. The statistical analysis in the results was evaluated by SPSS Statistics 17.0 and Prism 6.0.

**Figure 3 vaccines-10-00269-f003:**
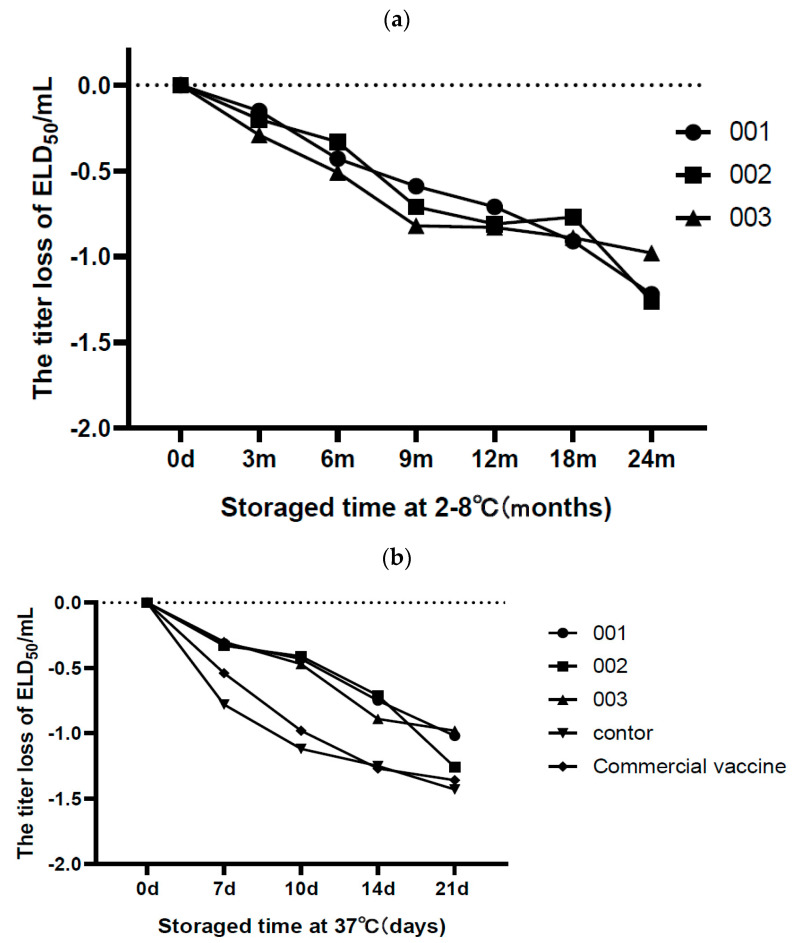
The thermal stability of the vaccine. A7 was tested at 2–8 °C (**a**); the A7 and commercial vaccines at 37 °C (**b**) and 45 °C (**c**) were tested in three lots (001, 002, and 003). The freeze-dried DHV vaccine without added stabilizer was assessed as the negative control group.

**Figure 4 vaccines-10-00269-f004:**
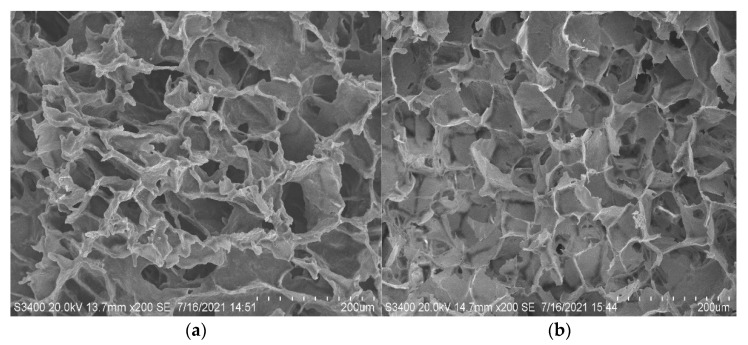
(**a**) SEM of commercial vaccines; (**b**) SEM of freeze-dried vaccines with formulation A7. The structure of the freeze-dried samples of the A7 protective agent is the same as that of commercial seedlings from SEM.

**Figure 5 vaccines-10-00269-f005:**
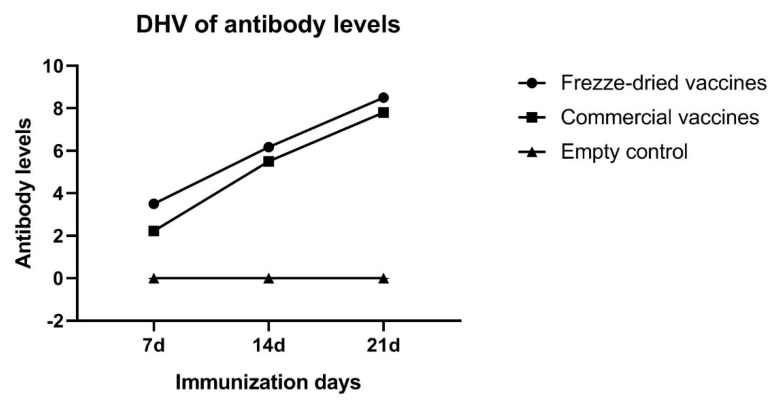
Antibody test results of the freeze-dried vaccines and commercial vaccines.

**Table 1 vaccines-10-00269-t001:** Lyophilized protection of the DHV formulation.

Formulation	Sucrose	Pullulan	PVP	Arginine
A1	5	1.2	0.5	1
A2	5	1.2	0.5	2
A3	5	1.2	0.5	1
A4	5	1.2	0.5	2
A5	10	1.2	0.5	1
A6	10	1.2	0.5	2
A7	10	1.2	0.5	1
A8	10	1.2	0.5	2

**Table 2 vaccines-10-00269-t002:** Grouping of DHV immunization test.

Serial Number	Group	Immunizing Dose	Storage Time at 2–8 ℃	Number of Ducks
1	Freeze-dried vaccines	1.0 ml	12months	10
2	Commercial vaccines	1.0 mL	12 months	10
3	No virus vaccines	1.0 normal saline	12 months	10

**Table 3 vaccines-10-00269-t003:** The thermal parameters of formulation A7.

Formulation	The Eutectic Point	Collapse Temperature	Glass Transition Temperature
A7	−27.3 °C	−38.4 °C	56.8 °C

**Table 4 vaccines-10-00269-t004:** Physical properties of the DHV vaccine and remaining moisture measurement results.

Vaccine Batch Lots	Physical Properties	Remaining Moisture (<4%, *n* = 4)	The Degree of Vacuum
001	Loose and spongy, easy off the wall, and dissolved rapidly after dilution; Redissolve rapidly	1.68 ± 0.08	Qualified
002		1.76 ± 0.02	Qualified
003		1.54 ± 0.06	Qualified

**Table 5 vaccines-10-00269-t005:** Grouping of animals in the safety trial for the heat-resistant freeze-dried protective agent.

Group	Number of Ducks	Immunization Route	Immunizing Dose	Number of Ducks Who Died	Weight Gain of Ducks (g)		
					Pre-Immune	14 Days after Immunization	Weight Gain
1	20	Subcutaneously inoculated	10 times	0/20	60.33 ± 3.72 a	565.17 ± 11.88 a	504.83 ± 8.16 a
2	20	Intramuscular inoculated	10 times	0/20	59.83 ± 3.02 a	563.67 ± 16.68 a	503.84 ± 13.75 a
Empty control group	20	Empty control	Not immunized	0/20	59.66 ± 3.32 a	565.33 ± 13.39 a	505.67 ± 10.07 a

a: Indicates that the difference is not significant.

## Data Availability

Not applicable.

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
