# Peer review of "Optimization of Heat-Resistance Technology for a Duck Hepatitis Lyophilized Live Vaccine"

_vaccines, 2022, doi:10.3390/vaccines10020269_

Round 1

Reviewer 1 Report

Thank you very much for allowing me to review the article entitled “Optimization of Heat-Resistance Technology for Duck Hepatitis Lyophilized Live Vaccine” (vaccines-1460188).

The thermolabile nature of commercially available Duck viral hepatitis (DHV)vaccines necessitate their storage, transportation, and dissemination under refrigerated conditions, these steps increase the final costs of vaccines.

In this conditions the aim of this study was to design a heat-resistant protective agent for sucrose,  lactose,  pullulan,  PVP,  sorbitol,  and  arginine,  according  to the  eutectic  point  of  the  formulation,  using  scientifically  determined  temperatures  in freeze drying procedures to accelerate an anti-aging test with 37 and 45 °C residual water for screening indicators.

At the end they found a good heat-resistant composition and freeze-dry-Ing process, that can be useful with a view to using suitable temperatures for animal vaccines for the development of new formulations and to provide technical support.

Comments:

Introduction: it´s well-structured allowing to identify the focus of the work and uses an adequate bibliography. The objective is clearly stated.

Material and methods: They used duck viral hepatitis type 1 DHV-JS strain was isolated in our laboratory (microbial preservation number CGMCC no. 8159) and duck embryos were used at 11–12 days of gestation, and ducklings were used at 1–7 days of age but They should specify the sample size used, and the sample size calculation they are based on should be included. (20 ducks, there is in results, but it should be in material and methods as well).

 The sections in which the methodology is subdivided are adequate and these sections are clearly described allowing to identify transparency in the experimentation.

Results: It´s structured in sections that allow a better understanding of the results.

Discussion: It´s well developed and allows the results obtained and the potential applications to be compared with the findings of other articles.

 Conclusion: the conclusions should be the contribution of the work, but there are written as if it were a summary. I also do not consider that the last sentence in line 327 to 329 is adequate since it is a subjective assessment by the authors. For all of which he considered that the conclusions should be rewritten.

Author Response

Dear editor,

Thank you for your comments and the extension in time for editing the paper. I have read and seriously considered all of the comments and revised the manuscript based on the reviewer’s recommendations. My responses to the reviewer’s concern are below.

Conclusion: the conclusions should be the contribution of the work, but there are written as if it were a summary. I also do not consider that the last sentence in line 327 to 329 is adequate since it is a subjective assessment by the authors. For all of which he considered that the conclusions should be rewritten.

 Answer: Thank the reviewers for affirming the manuscript. At the same time, the conclusions in the manuscript have been optimized.

Reviewer 2 Report

The manuscript “Optimization of Heat-Resistance Technology for Duck Hepatitis Lyophilized Live Vaccine” described the use of sucrose, lactose, pullulan, PVP, sorbitol, and arginine as temperature protective agent for duck hepatitis vaccine. Authors perform a series of measures, including EID50, Vaccine Heat-Stability Test, collapse Temperature, Eutectic Point, SEM and safety test in animals. The article needs a major revision in the English language.

Concerns:

  1. The authors claim to use sucrose, lactose, pullulan, PVP, sorbitol, and arginine. But only a few compounds are changed (two), and even these have only two variations each. What is the reason for this? In my opinion, in basic tests, changes in other compounds should be tested as well. This is the most critical aspect of the work, as the authors do not clarify why they do not use other combinations.

  1. It is extremely difficult to see differences in the picture generated by Scanning Electron Microscopy. Authors should demonstrate the differences more clearly.

  1. Did the group of animals that were not immunized receive a placebo? What was the immunization period? What is the interval between doses?

  1. Why did the animals not have biochemical and hematological parameters evaluated? Another parameter that should have been evaluated is movement and behavior. Measuring zootechnic parameters alone makes this section very subjective.

  1. Figure 2. Is the title reduction compared to who? This must appear in the description and figure.

Author Response

Dear editor,

Thank you for your comments and the extension in time for editing the paper. I have read and seriously considered all of the comments and revised the manuscript based on the reviewer’s recommendations. My responses to the reviewer’s concern are below.

1、The authors claim to use sucrose, lactose, pullulan, PVP, sorbitol, and arginine. But only a few compounds are changed (two), and even these have only two variations each. What is the reason for this? In my opinion, in basic tests, changes in other compounds should be tested as well. This is the most critical aspect of the work, as the authors do not clarify why they do not use other combinations.

Answer: Thank you for the good suggestion. We have supplemented the pullulan and the related role of PVP, and added the corresponding description in the revised supporting information (lines 337-344).

2、It is extremely difficult to see differences in the picture generated by Scanning Electron Microscopy. Authors should demonstrate the differences more clearly.

Answer: Thank you for the good suggestion. The purpose of the display in Figure 4 is to show that there is no difference in structure between the vaccine product developed by our process and the commercial vaccine on the market, and it is fully explained in the label.

3、Did the group of animals that were not immunized receive a placebo? What was the immunization period? What is the interval between doses?

The unimmunized ducks served as the experimental control group and did not receive a placebo. The ducks in the experimental group were immunized once and observed for 14 consecutive days with no dosing interval. This method refers to Deng's patent data [23].

4、Why did the animals not have biochemical and hematological parameters evaluated? Another parameter that should have been evaluated is movement and behavior. Measuring zootechnic parameters alone makes this section very subjective.

Answer: This safety experiment was carried out strictly and objectively in accordance with the "Regulations of the People's Republic of China on Biological Products"(See reference [26] for details), not subjective judgment, and biochemical and hematological parameters will be measured in future studies.

5、Figure 2. Is the title reduction compared to who? This must appear in the description and figure.

Answer: Thank you for the good suggestion. A title has been added to Figure 2 and has been revised in the manuscript.

Reviewer 3 Report

In this manuscript, Zhao et al. optimized freeze-drying process for duck hepatitis A virus genotype 1 (DHAV-1) vaccine candidate and evaluated the safety of DHAV-1 vaccine candidate. The data showed that the vaccine candidate with optimized heat protectors was allowed to be kept for long time, and that the vaccine was proved to be safe. In general, the results are sound, and support the conclusion. However, the quality of this manuscript needs to be improved. Specific comments are listed below.

  1. In instruction section, there is no study showing that DHAV can be divided into three serotypes as no cross-neutralization between DHAV-2 and DHAV-3 was performed. Please clarify.
  2. In this study, the authors evaluated the safety of freeze-dried vaccine. The authors need also proved the data about the antibody response after immunization. The protective efficacy of freeze-dried vaccine should be confirmed.
  3. The manuscript is not well-written. I believed that the authors need to polish the manuscript thoroughly.

Author Response

Response to Reviewer 3 Comments

Dear editor:

Thank you for your comments and the extension in time for editing the paper. I have read and seriously considered all of the comments and revised the manuscript based on the reviewer’s recommendations. My responses to the reviewer’s concern are below.

In this manuscript, Zhao et al. optimized freeze-drying process for duck hepatitis A virus genotype 1 (DHAV-1) vaccine candidate and evaluated the safety of DHAV-1 vaccine candidate. The data showed that the vaccine candidate with optimized heat protectors was allowed to be kept for long time, and that the vaccine was proved to be safe. In general, the results are sound, and support the conclusion. However, the quality of this manuscript needs to be improved. Specific comments are listed below.

 Point 1:In instruction section, there is no study showing that DHAV can be divided into three serotypes as no cross-neutralization between DHAV-2 and DHAV-3 was performed. Please clarify. 

Response 1: Thank the reviewers for affirming the manuscript, for Point 1 abou the types of duck hepatitis  in the cited literature 1 it is clear that duck hepatitis can be divided into three types.

  • Wu F, Lu F, Fan X, Pan Q, Zhao, S., Sun, H.; et al. Development of a live attenuated duck hepatitis A virus type 3 vaccine (strain SD70). Vaccine, 2020, 38, 4695–4703.

Point 2: In this study, the authors evaluated the safety of freeze-dried vaccine. The authors need also proved the data about the antibody response after immunization. The protective efficacy of freeze-dried vaccine should be confirmed.

Response 2: 

Immunogenicity test of DHV freeze-dried vaccines

The immunogenicity test was divided into 3 groups, 7-day-old ducklings in each group, subcutaneously injected into the neck, single immunization, blood was collected at 7d, 14d, and 21d after immunization, and the serum was separated. After mixing by volume, the duck embryo neutralization test was used to determine the level of duck viral hepatitis antibody in duck blood. The specific test groups are shown in Table 2.

Table 2. Grouping of DHV immunization test

Serial number

Group

Immunizing Dose

Storaged time at 2-8℃

Number of Ducks

1

Freeze-dried vaccines

1.0ml

12months

10

2

Commercial vaccines

1.0ml

12 months

10

3

No virus vaccines

1.0 normal saline

12 months

10

Figure 5. Antibody test results of the freeze-dried vaccines and commercial vaccines

 Point 3: The manuscript is not well-written. I believed that the authors need to polish the manuscript thoroughly.

Thank the reviewers for affirming the manuscript,for Point 3 abou The manuscript is not well-written. I want to say of my manuscript has been edited on the editing platform of the journal.

Round 2

Reviewer 2 Report

The article brings an extremely important theme. I welcome the efforts of the authors, but I do not see a significant methodological improvement in R1.

Author Response

Point 1: The authors claim to use sucrose, lactose, pullulan, PVP, sorbitol, and arginine. But only a few compounds are changed (two), and even these have only two variations each. What is the reason for this? In my opinion, in basic tests, changes in other compounds should be tested as well. This is the most critical aspect of the work, as the authors do not clarify why they do not use other combinations.

Response 1: Thank you for the good suggestion. We have supplemented the pullulan and the related role of PVP, and added the corresponding description in the revised supporting information (lines 337-344).

Point 2: It is extremely difficult to see differences in the picture generated by Scanning Electron Microscopy. Authors should demonstrate the differences more clearly.

Response 2: Thank you for the good suggestion. The purpose of the display in Figure 4 is to show that there is no difference in structure between the vaccine product developed by our process and the commercial vaccine on the market, and it is fully explained in the label.

Point 3: Did the group of animals that were not immunized receive a placebo? What was the immunization period? What is the interval between doses?

Response 3: The unimmunized ducks served as the experimental control group and did not receive a placebo. The ducks in the experimental group were immunized once and observed for 14 consecutive days with no dosing interval. This method refers to Deng's patent data [23].

Point 4: Why did the animals not have biochemical and hematological parameters evaluated? Another parameter that should have been evaluated is movement and behavior. Measuring zootechnic parameters alone makes this section very subjective.

Response 4: This safety experiment was carried out strictly and objectively in accordance with the "Regulations of the People's Republic of China on Biological Products"(See reference [26] for details), not subjective judgment, and biochemical and hematological parameters will be measured in future studies.

Point 5: Figure 2. Is the title reduction compared to who? This must appear in the description and figure.

Response 5: Thank you for the good suggestion. A title has been added to Figure 2 and has been revised in the manuscript.

Reviewer 3 Report

The manuscript is much improved, and I believed that it is ready for publication. Based on the article that the authors recommended (PMID: 32446833), there is no data validating the cross-neutralization activity between DHAV-2 and DHAV-3. I suggested to use “genotypes” instead of “serotypes”.